# Iron-Induced Oxidative Stress in Human Diseases

**DOI:** 10.3390/cells11142152

**Published:** 2022-07-08

**Authors:** Teruyuki Kawabata

**Affiliations:** Department of Applied Physics, Postgraduate School of Science, Okayama University of Science, Okayama 700-0005, Japan; kawabata@ous.ac.jp; Tel.: +81-86-256-9633

**Keywords:** iron, oxidative stress, redox-active iron, non-transferrin bound iron, transit iron pool, reactive oxygen species

## Abstract

Iron is responsible for the regulation of several cell functions. However, iron ions are catalytic and dangerous for cells, so the cells sequester such redox-active irons in the transport and storage proteins. In systemic iron overload and local pathological conditions, redox-active iron increases in the human body and induces oxidative stress through the formation of reactive oxygen species. Non-transferrin bound iron is a candidate for the redox-active iron in extracellular space. Cells take iron by the uptake machinery such as transferrin receptor and divalent metal transporter 1. These irons are delivered to places where they are needed by poly(rC)-binding proteins 1/2 and excess irons are stored in ferritin or released out of the cell by ferroportin 1. We can imagine transit iron pool in the cell from iron import to the export. Since the iron in the transit pool is another candidate for the redox-active iron, the size of the pool may be kept minimally. When a large amount of iron enters cells and overflows the capacity of iron binding proteins, the iron behaves as a redox-active iron in the cell. This review focuses on redox-active iron in extracellular and intracellular spaces through a biophysical and chemical point of view.

## 1. Introduction

Iron is a vital nutrient, the deficiency of which is responsible for several symptoms of anemia. Many people suffer from iron deficiency, especially children and young women [1]. Iron overload is responsible for hereditary as well as secondary hemochromatoses [2,3,4,5]. Even under normal conditions, iron may cause pathological damage depending on local conditions to produce redox-active iron. For example, free catalytic iron may be present in ischemic [6,7] or inflammatory lesions [8,9] and thus result in iron-induced oxidative stress because the pH in such lesions decreases [10], owing to which iron is solubilized as ions. Therefore, iron is the mastermind behind many systemic and local human diseases. Iron acts as a double-edged sword [11] and has to be strictly controlled and sequestered so as to not generate reactive oxygen species (ROS).

A dangerous form of iron in the blood that does not bind to transferrin is known as non-transferrin bound iron (NTBI). NTBIs increase in systemic iron overload and local pathological condition. This entity is still not well known but may induce ROS formation and cause oxidative stress in humans [2,12]. Another dangerous form of iron in the cell that does not bind to ferritin and other strongly chelating proteins exists in transit iron pool (TIP). This entity and the size are also still unknown but induce ROS and injury the cell. The size of the TIP may always change depending on the conditions of the cell by iron import, export and its usage.

Redox-active iron activates oxygen by the catalytic activity and produces ROS. In turn, the oxygen tension in the cell controls iron metabolism by hypoxia-inducible factors (HIFs). There is a significant connection between iron and oxygen. HIFs induced by hypoxia are transcription factors of transferrin and transferrin receptor. Conversely, iron regulatory proteins (IRPs) regulate HIF2α translation by iron-responsive element (IRE).

This review focuses on redox-active iron in extracellular and intracellular spaces by interdisciplinary approach.

## 2. Basic Properties of Iron

Iron is a 3d transition element containing five incompletely occupied d orbitals and has an electron configuration of [Ar]3d${}^{6}$4s${}^{2}$. In the human body, iron is usually present in two stable ionic states—ferric or ferrous.
$${\mathrm{Fe}}^{3+}+e⇄{\mathrm{Fe}}^{2+}$$

Ferric ions have an electron configuration of [Ar]3d${}^{5}$, with a ${}^{6}$S${}_{5/2}$ spectral term in the ground state as well as a completely symmetric structure. At its physiological pH 7, ferric ion is substantially insoluble and is deposited as ferric hydroxide colloids [13,14]. It is recommended that ferric ions be stored in strong acids, such as nitrate, in the laboratory. Researchers have often experienced that iron is deposited on the walls of test tubes containing an aged solution even if it is stored with some chelating ligand. To exist in the ionic state in human fluids, ferric ions need a chelating ligand. When ligands in a ferric ion coordinate, five symmetric d-orbitals split the ground energy into two energy levels–t${}_{2g}$ and e${}_{g}$–in a typical octahedral geometry. The energy difference Δ between t${}_{2g}$ and e${}_{g}$ is known as the crystal field splitting energy. Five electrons occupy each different orbital with the same upper spin in high-spin iron compounds, or the electrons occupy lower energy levels with electron pairs and only one electron has a half spin in the low-spin state. Iron bound to transferrin and many weak chelating ligands exist in ferric high-spin states because of the small Δ.

The solubility product of ferric ion is $10^{-39}$ M and [Fe${}^{3+}$] is 10${}^{-18}$ M at pH 7 [14]. According to the theory of hard and soft acids and bases, ferric ion is a hard acid that binds some hard bases, for example H${}_{2}$O and OH${}^{-}$. Ferric ion is insoluble in water and can form diiron, triiron, and polyiron coordination compounds. Not enough data on the reactivity of polyiron complexes in biology and medicine are available. Okazaki et al. [15] and Mizuno et al. [16] reported the role of diiron complexes in Fe-nta-induced renal injury and carcinogenesis. Fe-nta has a μ-oxo, μ-carbonato diiron complex [17] and it is suggested that the peroxide adduct of Fe-nta has a unique reactivity [18,19].

High-valent states of iron–Fe(IV), Fe(V), and Fe(VI)–have been reported in iron-containing enzymes and the model compounds. For example, iron may exist in the Fe(IV) transient state [20] in peroxidase reactions.

Ferrous ions have an [Ar]3d${}^{6}$ electron configuration, with a ${}^{5}$D${}_{4}$ spectral term, and typically possess tetrahedral or octahedral structures owing to ligand binding. Ferrous ions are much more soluble in water than ferric ions; however, they also easily become oxidized to ferric ions and deposit as iron colloids.

Fe^2+^ + O_2_ → Fe^2+^ − O_2_ ⇄ Fe^3+^ − O_2_^−·^ + Fe^2+^ → Fe^3+^ − O_2_ − Fe^3+^ → colloids

Colloidal ferric (hydr)oxide formation is a complex reaction. Ferrous ions are more stable in strong acids but are oxidized slowly in atmosphere, even when stored in acid.

The reduction potential of the half-cell couple Fe${}^{3+}$/Fe${}^{2+}$ is reportedly 0.771 V; however, the data mislead us into a wrong decision. The reduction reaction can be more precisely described as follows: $${\mathrm{Fe}}^{3+}{\left(H_{2}O\right)}_{6}+e⇄{\mathrm{Fe}}^{2+}{\left(H_{2}O\right)}_{6}.$$

The reaction occurs under ideal acidic conditions of pH 0; in more alkaline conditions, the water is displaced by hydroxyl ions. In biochemistry, it is customary to express the reduction potential at pH 7.

Furthermore, the use of reduction potential is recommended in deciding the possibility of a reaction. Let us postulate the redox reaction Ox1 + Rd2 → Rd1 + Ox2. We can calculate standard Gibbs energy, $\Delta G^{0}$, from the difference in the standard reduction potential, $\Delta \xi^{0}$;
$$\Delta G^{0}=-nF\Delta \xi^{0},$$
where *n* is the number of electrons and *F* is the Faraday constant. In the actual reaction, the change in Gibbs energy is as follows: $$\Delta G=\Delta G^{0}+RT\ln\frac{\left[\mathrm{Rd}1][\mathrm{Ox}2\right]}{\left[\mathrm{Ox}1][\mathrm{Rd}2\right]},$$
which indicates that for ΔG<0, the possibility of the reaction depends on not only $\Delta G^{0}$ but also on the concentration of each reactant and product. Therefore, if the change in Gibbs energy is slightly negative, the negative value is canceled by the reaction quotient and the reaction does not occur.

Table 1 summarizes the reduction potentials of some iron chelates. The reduction potentials of the iron coordination compounds dramatically change from negative to positive values.

Ferrous ions are hazardous to cells and produce oxidative stress in humans. In a well-known Fenton reaction, ferrous ions are oxidized by H${}_{2}$O${}_{2}$ and generate hydroxy radicals as follows: (1)$${\mathrm{Fe}}^{2+}+H_{2}O_{2}\to {\mathrm{Fe}}^{3+}+{\mathrm{OH}}^{-}+\cdot \mathrm{OH}.$$

In the presence of some reductant such as superoxide,
(2)$${\mathrm{Fe}}^{3+}+O_{2}^{-\cdot }\to {\mathrm{Fe}}^{2+}+O_{2}.$$

By adding two chemical Equations (Equations (Equation 1) and (Equation 2)), we obtain the following: (3)$$O_{2}^{-\cdot }+H_{2}O_{2}\to O_{2}+\cdot \mathrm{OH}+{\mathrm{OH}}^{-},$$
which is well known as the Haber–Weiss reaction [23]; however, Equation (Equation 3) cannot proceed without a catalytic iron.

A solution of ferrous irons and hydrogen peroxide has been used as the Fenton reagent to oxidize organic molecules. However, the actual mechanism underlying the Fenton reaction remains controversial [24]; it is unknown whether free hydroxy radicals are generated or if a coordination complex of iron and hydrogen peroxide [for example, Fe(IV)=O] attack organic molecules [25]. It is also possible that iron binds weakly to an organic molecule, after which hydrogen peroxide reacts at the site. The detailed mechanism may depend on the nature of the oxidized molecules and the reaction environments.

In the fields of biology and medicine, the spin trapping method has often been used to identify ROS in concerned systems including experiments on iron-induced oxidative stress. 5,5-Dimethyl-1-pyrroline-N-oxide (DMPO) has been frequently used as a spin trap because the spin adducts to superoxide and hydroxy radicals, DMPO-OOH and DMPO-OH (Figure 1), can be identified at a glance. A simulation of the DMPO spin adducts was performed using EasySpin software [26]. Spin trapping studies using DMPO have been performed to identify oxygen-derived free radicals in iron-induced oxidative stress, and DOMP-OH adducts have been reported as a direct evidence of the presence of hydroxy radical [27]. However, one must be cautious of DMPO spin trapping in iron-rich systems [28,29,30]. DMPO can chelate ferric ions, and non-specific DMPO adducts can be generated in iron-containing systems.

## 3. Redox-Active Iron in Blood

In the human blood and in extracellular fluid, iron is principally bound to serum transferrin, which contains a ∼79-kD glycoprotein and binds two ferric ions per molecule: (4)$$\begin{array}{c} {\mathrm{Fe}}^{3+}+\mathrm{Tf}⇄\mathrm{FeTf},\;K_{1}=\left[\mathrm{FeTf}\right]/\left[\mathrm{Fe}\right]\left[\mathrm{Tf}\right] \end{array}$$(5)$$\begin{array}{c} {\mathrm{Fe}}^{3+}+\mathrm{FeTf}⇄{\mathrm{Fe}}_{2}\mathrm{Tf},\;K_{2}=\left[{\mathrm{Fe}}_{2}\mathrm{Tf}\right]/\left[\mathrm{Fe}\right]\left[\mathrm{FeTF}\right]. \end{array}$$

The stability constants K${}_{1}$ and K${}_{2}$ are very large, and almost all iron is bound to transferrin in the blood. However, several ligands can chelate serum iron:

Fe^3+^ + L ⇄ FeL.

These ligands may be low molecular weight ligands or macromolecules such as proteins other than serum transferrin. Because ferric ion is a hard acid, hard bases such as acetate and phosphate groups may bind the iron. Polyiron coordination compounds may also be formed with different ratios of iron to ligands. Some of these iron coordination compounds are possibly redox-active and give rise to oxidative stress in the human body.

### 3.1. Serum Transferrin

Apo-transferrin has been used to suppress iron-induced oxidative stress in the experimental systems. The structure and properties may suggest the nature of non-catalytic iron to us.

Serum transferrin is the principal iron transporter in the blood; it binds one iron in the N-lobe and another in the C-lobe (Figure 2) and maintains ferric ion solubility in the extracellular space. Serum transferrin is mainly synthesized in the hepatocytes and secreted into the blood after protein modifications such as glycosylation and phosphorylation [31]. Transferrin is known to bind other metal ions such as Ti(III), Ga(III), and Co(II).

The structures of iron-bound and non-bound transferrin were elucidated using crystallographic analyses and are available in the Protein Data Bank (PDB). Both sites have two tyrosines, aspartate, histidine, and carbonate (a synergistic ion) (Figure 3).

The iron binding reaction is as follows:

Fe^3+^ + H_3_ − Tf + CO_3_^2−^ ⇄ Fe^3+^ − Tf − CO_3_^2−^ + 3H^+^.

The three protons come from two tyrosines and an aspartate. Iron transferrin has a salmon pink color that is ascribed to ligand-to-metal charge transfer from tyrosine to iron. Apart from carbonate, iron transferrins coordinated by oxalate and citrate are available in PDB, indicating that oxalate and citrate may bind iron coordinated by transferrin in blood.

The mechanism underlying iron capture by transferrin remains unclear [31]. The first step is the binding of carbonate, after which iron is inserted to the site and, finally, the cleft of the lobe is closed.

The iron-binding sites between the N- and C-lobes have different properties although the iron in both sites is coordinated by the same amino acids and carbonate (Figure 3). The stability constants are given in Table 2. The constant for the N-site is greater than that for the C-site, and the both constants decrease under acidic conditions than under alkaline conditions. In Table 2, k${}_{1N}$ and k${}_{1N}$ are the microscopic stability constants of one iron binding without iron in another site, and k${}_{2N}$ and k${}_{2N}$ are the stability constants of the second iron binding with another site occupied by iron. However, in serum, iron occupies the C-site more frequently than the N-site, the reason for which remains unclear [31].

The magnetic properties of the iron in transferrin have been studied using electron paramagnetic resonance. Electron paramagnetic resonance (EPR) spectra also support slightly (but not drastically) different properties between the N- and C-sites. Recently, EPR studies of iron transferrin at high frequency microwaves (275 GHz) [34,35] have been reported. For analysis, the fourth-order term, $\sum B_{4}^{q}O_{4}^{q}$ (q = −4, ⋯, 4), is added to the usual spin Hamiltonian Equation [36]: $$H=\mu_{B}B\cdot g\cdot S+D\left(S^{2}-S_{z}^{2}\right)+E\left(S_{x}^{2}-S_{y}^{2}\right),$$
where g is the g-tensor; ***B*** is the magnetic field; ***S*** is the effective spin operator; and $S_{x}$, $S_{y}$, and $S_{z}$ are the *x*, *y*, and *z* elements, respectively. *D* is a zero-field splitting parameter, whereas E/D is the rhombicity. Compared with that in an usual X-band EPR study, in a high-field EPR study, two irons in the N- and C-lobes have been differentiated much more clearly. Iron transferrin exhibits a characteristic double peak at g∼4.3 through the use of X-band EPR, the reason for which has been clearly explained by the fourth-order term. The D value was also directly estimated through the simulation.

### 3.2. A Candidate for Redox-Active Iron in Blood: NTBI

Hershko et al. [37] reported the presence of non-specific serum iron in thalassemia; since then, the amount of iron chelatable by sufficient amounts of transferrin has been investigated. NTBI literally refers to iron in extracellular fluid, especially in the blood, that is not bound to serum transferrin. This includes several iron compounds that do not bind transferrin. In terms of iron-induced oxidative stress, NTBI is a possible candidate; however, not all NTBIs are injurious to cells.

Some other terms such as “free” and “labile” are used in the literature. At the physiological pH, ferric ions cannot exist as free ions in aqueous solution, and labile iron ligands indicate that the ligand is rapidly exchangeable in coordination chemistry fields. The two terms may be confusing. In this review, we were more interested in redox-active NTBIs that can induce oxidative stress.

Such NTBIs may be bound by small ligands that can chelate iron (low molecular weight iron); the same may be applicable to macromolecules as well. For example, it is well known that serum albumin binds copper. Serum ferritin is clearly such a macromolecule, but iron packed in ferritin is known to be non-redox-active. Many researchers believe that such iron molecules can be catalytic for ROS generation and induce oxidative stress. However, the actual entities still remain unclear. A possible candidate is iron citrate. Ferric ion is a hard acid and can be coordinated by acetate, phosphate, carbonate, nitrate, and hydroxylate groups as well as other hard bases.

### 3.3. Estimation of NTBI in the Presence of Ligands

Estimation of NTBI in the presence of a ligand may provide a useful guide for the experimental assay. Ferric ions are a typical ligand-labile transition metal ion and rapidly exchange ligands. It is reasonable that iron and transferrin are in equilibrium in the blood. Using Equation (Equation 4),
$$K_{1}=\frac{\left[\mathrm{FeTf}\right]}{\left[\mathrm{Fe}][\mathrm{Tf}\right]},$$
where [X] denotes the concentration of X. Then,
(6)$$\left[\mathrm{FeTf}\right]=K_{1}\left[\mathrm{Fe}\right]\left[\mathrm{Tf}\right].$$

Likewise, from Equations (Equation 5) and (Equation 6),
(7)$$\left[{\mathrm{Fe}}_{2}\mathrm{Tf}\right]=K_{2}\left[\mathrm{Fe}\right]\left[\mathrm{FeTf}\right]=K_{1}K_{2}{\left[\mathrm{Fe}\right]}^{2}\left[\mathrm{Tf}\right].$$

Therefore, total concentration of Tf (${\mathrm{Tf}}_{0}$)
$$\begin{array}{cc} {\mathrm{Tf}}_{0} & =\left[\mathrm{Tf}\right]+\left[\mathrm{FeTf}\right]+\left[{\mathrm{Fe}}_{2}\mathrm{Tf}\right]\\ \; & =\left[\mathrm{Tf}\right]+K_{1}\left[\mathrm{Fe}\right]\left[\mathrm{Tf}\right]+K_{1}K_{2}{\left[\mathrm{Fe}\right]}^{2}\left[\mathrm{Tf}\right]\\ \; & =\left[\mathrm{Tf}\right](1+K_{1}\left[\mathrm{Fe}\right]+K_{1}K_{2}{\left[\mathrm{Fe}\right]}^{2}) \end{array}$$

Then, we can calculate the concentration of free Tf,
$$\left[\mathrm{Tf}\right]=\frac{{\mathrm{Tf}}_{0}}{1+K_{1}\left[\mathrm{Fe}\right]+K_{1}K_{2}{\left[\mathrm{Fe}\right]}^{2}}.$$

If we know the free iron concentration [Fe], we can decipher the unknown iron species from Equations (Equation 6) and (Equation 7).

When a ligand for ferric ion are added to blood, we need the following extra equation: $${\mathrm{Fe}}^{3+}+L⇄\mathrm{FeL}$$

From the chemical equilibrium, the concentration of a free ligand is
$$\left[L\right]=\frac{L_{0}}{1+K[\mathrm{Fe}]},$$
where L${}_{0}$ is the total concentration of the ligand, L, and K is its stability constant. If we know the free iron concentration [Fe], we can estimate the concentration of the iron coordination compound: $$\left[\mathrm{FeL}\right]=K\left[\mathrm{Fe}\right]\left[L\right]=K\frac{L_{0}\left[\mathrm{Fe}\right]}{1+K[\mathrm{Fe}]}.$$

Therefore, in a system involving ferric ions, Tf, and ligands, we can calculate all species of iron complexes if the concentration of free iron [Fe] is known.

Figure 4a shows the pointer function [38] in a system involving free transferrin, FeTf, and Fe${}_{2}$Tf by changing the free iron concentration, in which it is supposed that the total transferrin concentration is 27.0 μM and the total iron concentration is 18.0 μM, both of which are normal in healthy adults. Figure 4b shows the pointer function in terms of the coexistence of transferrin and a ligand (500 μM), the stability constant of which is $10^{15.6}$. To compare the estimated data with the experimental assay, the stability constant of nta is used for the calculation. The point at which $\log_{10}\left|D\right|$ becomes minimal indicates the actual equilibrium state.

The concentrations of each iron coordination compound at equilibrium are summarized in Table 3. The stability constant of transferrin is approximately $10^{3}$–$10^{4}$ times larger than that of the ligand; thus, the concentration of iron transferrin is approximately the same between the both columns (without and with ligand).

The concentration of FeL equivalent of NTBI is different from that reported in the experimental data from Ito et al. [39] by one order of magnitude. If the ligand has a not-so-large stability constant, the presence of the ligand cannot disturb NTBI assays. However, under conditions wherein FeL is removed from the assay system, we cannot ignore the effect of the ligand.

### 3.4. Experimental Assay of NTBI

Halliwell and Gutteridge developed a method to detect redox-active iron as a catalytic iron [12,40]. They measured NTBI as bleomycin-detectable iron. Iron chelated by bleomycin produces TBAR with ascorbate and DNA; TBAR was measured using UV-vis spectroscopy. The method is good and reliable and does not detect NTBI in normal blood and body fluid. Now, the modified method is conducted using microplates [41].

After the bleomycin-detectable iron assay, other iron chelators such as DFO and nta were used in the measurement of NTBI. To separate chelatable iron from transferrin-bound iron, several filtration techniques were developed [12]. Next, we used some chelators, which remove iron from iron transferrin depending on the treatment times: Fe2-Tf + L → Fe-Tf + FeL. If the amount of FeL cannot be neglected as compared with true NTBI, the NTBI values may be overestimated.

Recently, Ito et al. [39] developed an automated assay method applicable in clinical practice. They reported that NTBI was 0.44 ± 0.076 μM in healthy volunteers with no difference in both sexes.

An electrochemical method was recently reported by Angoro et al. [42]; this method may be promising in future clinical practice.

## 4. Redox-Active Iron in Cells: Transit Iron Pool

Although redox-active iron in cells is dangerous and damages the cells by generating ROS, cells definitely need iron for cell homeostasis. For cells to use such dangerous iron for survival, iron is finely controlled and carefully treated via binding to some proteins. One can imagine an iron pool in transient migration from one functional macromolecule to another. This iron pool is referred to as a labile iron pool, but as already mentioned, the term “labile” is slightly confusing. Such iron pools have been designated as TIP in this review.

### 4.1. Iron Balance in Cells

The mechanism underlying iron uptake and release in cells has been briefly summarized in this section. Iron is ubiquitous and necessary for all human cell functions. Iron transferrin in serum binds transferrin receptors (TFR1 and TFR2) and is transported into microsomes [43]. TFR1 is present in almost all cells, but TFR2 is primarily present in only hepatocytes and erythroid progenitor cells. Microsomes are acidified by the action of proton pumps, and iron is released from transferrin. This iron is reduced by ferrireductases such as six-transmembrane epithelial antigen of prostate 3 (STEAP3) and transported out of the microsomes by DMT1 [44]. Small amounts of NTBI may be taken up by ZRT/IRT-like protein 14 (ZIP14), L-type and T-type calcium channels, ZIP8, and transient receptor potential cation channel subfamily C member 6 (TRPC6) [45]. DMT1 transports the iron to PCBP1/2 [44,46,47,48,49,50], whereas PCBP1/2 carries the iron to other necessary places. Excess iron is brought to FPN1 and excreted into the extracellular space [51,52]. The excreted iron is oxidized by hephaestin and transported to transferrin [53]. Iron-released transferrin is then transported to the cell membrane again and recycled to the extracellular space.

### 4.2. Ferritin: A Defence Molecule against Redox-Active Iron

Ferritin preserves iron in a safe state; thus, it is a direct defense molecule against redox-active iron [54]. It not only maintains the iron redox-inactive state in the protein cage but also is secreted to the extracellular space by iron-replete cells encapsulated in CD63-positive vesicles [55]. However, if iron is deficient in a cell, it is supplied from ferritin to TIP. In such conditions, ferritin could be a bomb and serve as a supplier of redox-active iron [56].

Ferritin comprises two different kinds of subunits—H-subunit and L-subunit—and is composed of a total of 24 subunits [57,58]. These subunits form a spherical cage and have a cavity in the protein structure, where ferric ions deposit a maximum of 4500 iron atoms per ferritin molecule. The ratio of the H-subunit to L-subunit is tissue-specific. The H-subunit has ferroxidase activity and oxidizes ferrous ions to ferric ions, whereas the L-subunit serves as a nucleation site for the iron. By the coordination of the H- and L-subunits, iron is stored in the core of ferritin.

Serum ferritin is a useful marker for iron storage [59] and the treatment or recurrence assessment of cancers [60]. It is primarily composed of the L-subunit, and the iron content is poor. It remains unknown how serum ferritin is obtained; however, it has been reported to be secreted through two different vesicular pathways [61].

### 4.3. Indicators for TIP: IRP1/2 and HIF-PHD

IRP1/2 and HIF-PHD are possible indicators for TIP, which may be useful for the assessment of TIP. IRP1/2 [62,63] regulates the translation of iron-related proteins such as transferrin and ferritin by binding iron-responsive elements of those mRNAs. Iron controls the binding and release of the mRNAs to IRE. When IRP1/2 binds iron, the proteins cannot bind the IREs of the mRNAs.

mRNA-IRP1 + Fe ⇄ mRNA + Fe-IRP1

Therefore iron-IRP1/2 may be a good indicator of TIP [12]. In particular, iron-IRP1 possesses aconitase activity, and the activity of the cytosolic aconitase may be a good marker of TIP.

Iron is a key atom in the activation of oxygen, and oxygen in turn regulates iron homeostasis in cells [64]. HIFs act as transcription factors for transferrin and transferrin receptors. Conversely, IRP controls the translation of HIF2α. When enough oxygen is available, HIFαs is degraded by proteasomes after hydroxylation. The hydroxylation is executed by HIF-prolyl hydroxylase domain (PHD), which needs iron as a cofactor. The hydroxylase activity can reflect the implications of TIP, such as the aconitase activity of iron-IRP1.

### 4.4. DMT1 and FPN1

TIP is regulated by the import and export of iron in the cell and the main molecules for the regulation, DMT1 and FPN1, are discussed here. Gunshin et al. reported that DMT1 is an important importer of iron to cytosol [65]. DMT1 is crucial for iron uptake in duodenal epithelia and other cells. The expression of DMT1 isoforms is dependent on the cell type and subcellular distribution. The protein structure of human DMT1 is not available in PDB; however, the homology modeling has been reported [66]. The precise molecular mechanism underlying iron transport remains unclear.

FPN1 is the only exporter of iron from cells, and hepcidin binding controls protein degradation. Taniguchi et al. elucidated the inward- and outward-facing structures of a bacterial homolog of FPN1; a hepcidin-binding site was also indicated [67]. FPN1 has two iron-binding sites: C-lobe and N-lobe. Hepcidin binds the upward-open structure and contacts the single iron-binding site [68,69].

The export and import of iron by these membrane proteins can be understood using a four-state model [70]. The transporter has an open structure that binds iron at the outside of the cell; next, the conformation changes and the open structure orients inward, toward the cell. The iron in the inward open site is then released into the cytosol (Figure 5).

Recently, a mathematical model of iron uptake by DMT1 in Caco2 cells was reported [71]; the model includes endocytosis, which is also included in the four-state model. The study reported two different models on endocytosis, the binary switching-mechanism model and the swinging-mechanism model, the latter being better for experiments.

### 4.5. Diffusion of Redox-Active Iron

In the experiments of iron-induced oxidative stress, some redox-active iron is added to cell culture media. To discuss the kinetics how such redox-active iron diffuses in cells may be important to understand the mechanism of iron-induced oxidative stress in the cell.

Iron imported into the cytosol is directly transferred to PCRP1/2, and the iron moves to specific sites with PCRP1/2 [44,46,47,48,49,50]. When cells are exposed to excess iron under iron overload conditions, the cytotoxic iron that enters into the cytosol may diffuse in the cell. In such cases, the transmission of the iron pool increases rapidly and the capacity of ferritin is overwhelmed. These possible conditions may occur in ischemic injury and other events related to cell death. Recently, the diffusion of proteins in cells has been directly measured through the assessment of fluorescence recovery after photobleaching (FRAP) [72,73], although the intracellular flow of cells has been reported [74] and specific proteins are transported by the transport system. Recently, the diffusion coefficient has been reported to be fluctuating depending on protein fluctuations [75]. This may help understand oxidative injuries caused by redox-active iron to simulate the diffusion of such iron in cells.

The diffusion of a particle can be described using the diffusion equation: $$\frac{\partial c}{\partial t}=D\nabla^{2}c,$$
where *c* is the concentration of a particle and *D* is the diffusion coefficient that depends on temperature *T* and the reciprocal radius of the particle, *r*: $$D=\frac{k_{B}T}{6\pi \eta r},$$
where $k_{B}$ is the Boltzmann constant and η is the viscosity of the fluid. The radius of the Stokes–Einstein equation for proteins was modified as described by Yamamoto et al. [75].

The diffusion coefficient of a fluorescent protein-EYFP-was reported to be ∼50 $\mu m^{2}$/s [76]. Excess iron in cells may be coordinated by small molecules; we assume *D* = 100 μm${}^{2}$/s, supposing that the molecular weight is proportional to the cubed radius of the molecule. The diffusion was simulated considering the cells are in an environment in which the iron concentration is constant (100 μM) (Figure 6).

The time required for the arrival of an iron molecule to the centre of the cell (distance = 25 μm) is ∼3 s from the equation $\langle x^{2}\rangle =2Dt$. If the iron is coordinated with some large protein, it takes several tens of seconds.

### 4.6. Experimental Assay of the Transit Iron Pool

Little is known about transit iron pools in cells, how much iron is present in TIP, and how iron exists in the cells. To elucidate the presence of TIP, several iron-chelatable fluorescent agents have been developed [77,78,79]. Some fluorescent agents often bind other ions and metals, and their specificity to ferric and ferrous ions is dicey. These fluorescent agents are quenched by iron binding, and it may be difficult to use them in some experiments. Hirayama et al. [80,81] developed a new fluorescent probe that is non-fluorescent but becomes fluorescent after reaction with ferrous ions; this fluorescence was applied to the clinical tissue sections [82]. They also developed several fluorescent agents to target specific organelles in the cells [83,84,85]. However, the fluorescent sites do not always indicate the presence of TIP; moreover, after the reaction of the fluorescent probes with iron, the reagents may accumulate in different cell compartments.

A new fluorescent probe that uses the principle of Fluorescence resonance energy transfer (FRET) has been reported [86]; this probe is very useful and convenient for practical experiments. It has two fluorophores, of which the first has a red color. After the cleavage of one fluorophore through reaction with ferrous ions, the fluorescence changes to green. When conducting experiments, we can check the position of the probe first and then confirm the position of the green-colored probe at which it react with ferrous ions. However, there exists some doubt as to what will the outcome be if the green fluorescent probe is replaced at another site different from the reaction site.

## 5. Conclusions

Iron plays a pivotal role in the generation of ROS, except in special conditions. Iron is ubiquitous in extracellular and intracellular spaces. Not all states of iron produce ROS; however, at any time, redox-active iron may appear, depending on the conditions. Therefore, it can be stated that iron is the mastermind behind several human diseases and plays roles in their etiology and pathogenesis.

## Figures and Tables

**Figure 1 cells-11-02152-f001:**
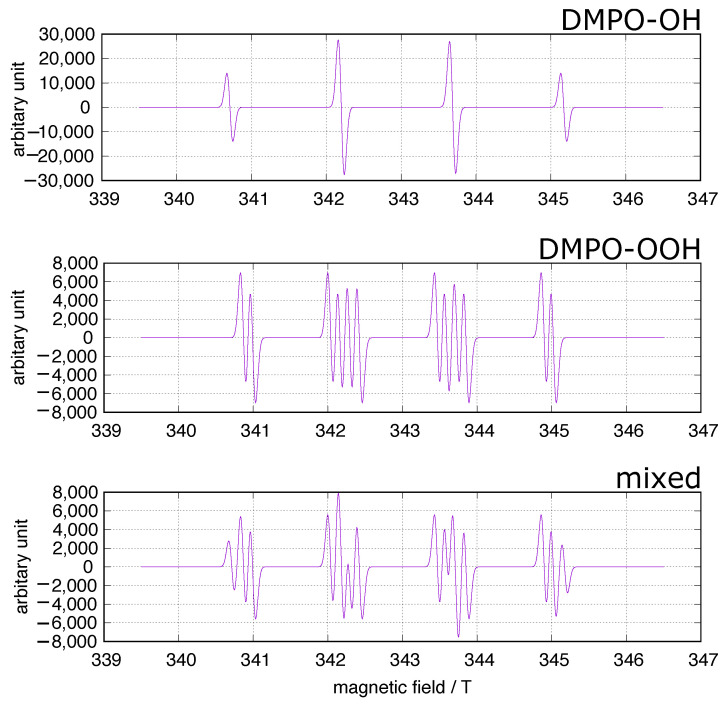
DMPO spin trapping. The simulation was performed using EasySpin software [26]. upper: DMPO-OH, hyperfine splitting constants (mT): A${}_{N}$ = 1.49, A${}_{H}$ = 1.48; middle: DMPO-OOH, hyperfine splitting constants (mT): A${}_{N}$ = 1.43, A${}_{H}^{\beta }$ = 1.17, A${}_{H}^{\gamma }$ = 0.125; lower: mixed DMPO-OH and DMPO-OOH.

**Figure 2 cells-11-02152-f002:**
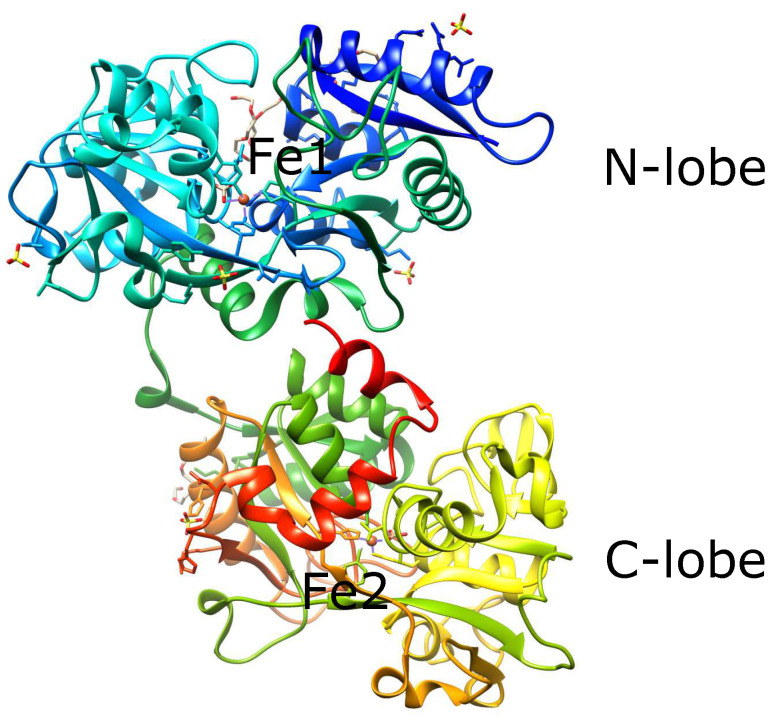
Human serum holotransferrin. The N-lobe is bluish and the C-lobe is lighter. Brown spheres in each lobe are iron ions. The protein structure was generated using UCSF Chimera [32]. PDB: 3V83.

**Figure 3 cells-11-02152-f003:**
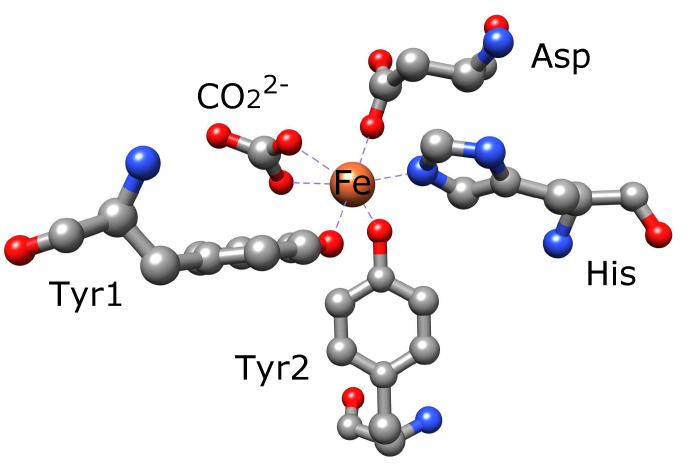
Iron-binding site of the N-lobe in human serum holotransferrin. Iron is coordinated by two tyrosine, histidine, aspartate, and carbonate. The iron in the C-lobe is also coordinated by the same amino acids. The structure was generated using UCSF Chimera [32]. PDB: 3V83.

**Figure 4 cells-11-02152-f004:**
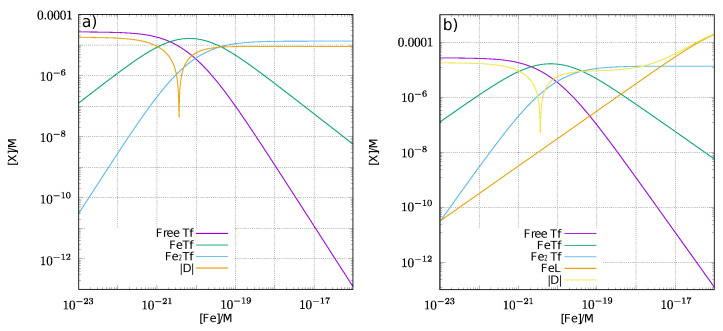
The dependence of each coordination species on the free iron concentration, [Fe]. The vertical line shows the log of each iron species, whereas the horizontal one shows the log of [Fe]. D is the difference in total iron from the sum of the iron of each species including free iron. The minimum point of $\log_{10}\left|D\right|$ indicates the actual equilibrium of each iron coordination compound. (**a**) without ligand; (**b**) with ligand, L.

**Figure 5 cells-11-02152-f005:**
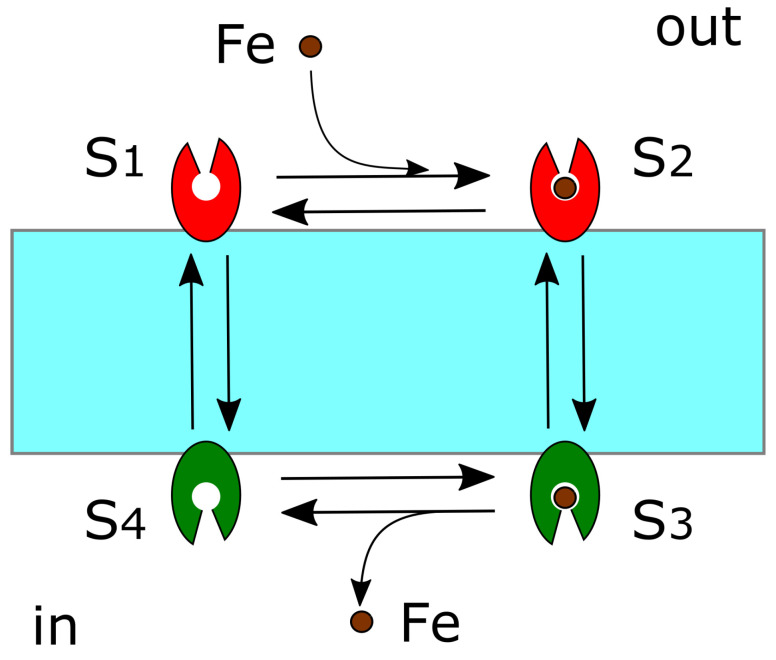
Four-state model. S1: The initial state opens outward; S2: The second state opens outward; S3: The third state turns inward; S4: The fourth state opens inward. In this figure, the four states are all reversible.

**Figure 6 cells-11-02152-f006:**
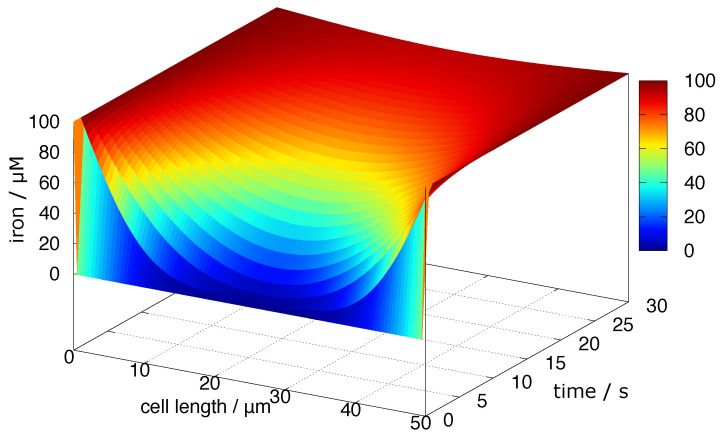
Diffusion of iron coordination compounds in the cell. The iron enters from both sides of the cell and diffuses in the cytosol. Cell size: 50 μm, *D* = 100 μm${}^{2}$/s.

**Table 1 cells-11-02152-t001:** Standard reduction potentials for some iron coordinate compounds. (1) ref [21]; (2) ref [22]; (3) ref [12].

Redox Couple	Reduction Potential
Fe${}^{3+}$/Fe${}^{2+}$	0.771 ${}^{\left(1\right)}$
Fe(OH)${}_{3}$/Fe(OH)${}_{2}$	−0.56 ${}^{\left(1\right)}$
[Fe(bipy)${}_{3}^{3+}]$/[Fe(bipy)${}_{3}^{2+}]$	1.03 ${}^{\left(1\right)}$
[Fe(phen)${}_{3}^{3+}]$/[Fe(phen)${}_{3}^{2+}]$	1.147 ${}^{\left(1\right)}$
Fe${}^{3+}$-nta/Fe${}^{2+}$-nta	0.59 ${}^{\left(2\right)}$
Fe${}^{3+}$-dtpa/Fe${}^{2+}$-dtpa	0.03 ${}^{\left(3\right)}$
Fe${}^{3+}$-edta/Fe${}^{2+}$-edta	0.12 ${}^{\left(3\right)}$
Fe${}^{3+}$-ADP/Fe${}^{2+}$-ADP	0.10 ${}^{\left(3\right)}$
Fe${}^{3+}$-DFO/Fe${}^{2+}$-DFO	−0.45 ${}^{\left(3\right)}$
Fe${}^{3+}$-Tf/Fe${}^{2+}$-Tf	−0.40 ${}^{\left(3\right)}$

**Table 2 cells-11-02152-t002:** Effective stability constants, macroscopic (K${}_{1}$, K${}_{2}$) and microscopic (k${}_{1C}$, k${}_{1N}$, k${}_{2C}$, k${}_{2N}$), for iron binding to transferrin (modified from [33]).

pH	K${}_{1}$/M		K${}_{2}$/M	
6.7	$3.0\times 10^{19}$		2.3 × 10${}^{17}$	
	$k_{1C}=2.9\times 10^{19}$	$k_{1N}<1.4\times 10^{18}$	$k_{2C}=4.8\times 10^{18}$	$k_{2N}<2.4\times 10^{18}$
7.4	$4.7\times 10^{20}$		$2.4\times 10^{19}$	
	$k_{1C}=4.0\times 10^{20}$	$k_{1N}=6.8\times 10^{19}$	$k_{2C}=1.6\times 10^{20}$	$k_{2N}=2.8\times 10^{19}$

**Table 3 cells-11-02152-t003:** Calculated iron coordination species in solution. Total Fe: 18.0 μM; total Tf: 27.0 μM; total L: 500 μM.

Species	without L/μM	with L/μM
free Fe	3.245 × 10${}^{-15}$	3.242 × 10${}^{-15}$
free Tf	10.21	10.22
FeTf	15.57	15.57
Fe${}_{2}$Tf	1.213	1.211
Free L	0	500.0
FeL	0	0.006453

## Data Availability

Publicly available datasets were analyzed in this study. This data can be found here: https://www.rcsb.org, accessed on 20 May 2022.

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
