# Peer review of "Iron-Induced Oxidative Stress in Human Diseases"

_cells, 2022, doi:10.3390/cells11142152_

Round 1
Reviewer 1 Report
Comments:
The author aimed to clarify the relation of iron in extracellular and intracellular spaces with generation of ROS. Detailed descriptions and rich charts are used to support the author's views and theories. The overall logic of this article is clear. However, there are still some problems that need to be modified.
Q1: The title of this manuscript was “Iron-induced oxidative stress: the mastermind behind human diseases.”, but the main point in your context was to introduce iron and its transition biophysical and chemical aspect. It seems to be abstract and unintelligible. I think the role and mechanism of iron - induced oxidative stress in disease should be added.
Q2: A detail: the citation format of reference should be checked and modified. For example, the citation in line 293 is not shown.
Q3: The relationship between iron and ROS generation in this manuscript should be more specific.
Reviewer 2 Report
see attached file

Round 2
Reviewer 2 Report
see attached
